# Chloride ions evoke taste sensations by binding to the extracellular ligand-binding domain of sweet/umami taste receptors

Nanako Atsumi[1†], Keiko Yasumatsu[1,2,3†], Yuriko Takashina[4], Chiaki Ito[1], Norihisa Yasui[1,4], Robert F Margolskee[3], Atsuko Yamashita[1,4]*

[1]Graduate School of Medicine, Dentistry and Pharmaceutical Sciences, Okayama University, Okayama, Japan; [2]Tokyo Dental Junior College, Tokyo, Japan; [3]Monell Chemical Senses Center, Philadelphia, United States; [4]School of Pharmaceutical Sciences, Okayama University, Okayama, Japan

**Abstract** Salt taste sensation is multifaceted: NaCl at low or high concentrations is preferably or aversively perceived through distinct pathways. $Cl^-$ is thought to participate in taste sensation through an unknown mechanism. Here, we describe $Cl^-$ ion binding and the response of taste receptor type 1 (T1r), a receptor family composing sweet/umami receptors. The T1r2a/T1r3 heterodimer from the medaka fish, currently the sole T1r amenable to structural analyses, exhibited a specific $Cl^-$ binding in the vicinity of the amino-acid-binding site in the ligand-binding domain (LBD) of T1r3, which is likely conserved across species, including human T1r3. The $Cl^-$ binding induced a conformational change in T1r2a/T1r3LBD at sub- to low-mM concentrations, similar to canonical taste substances. Furthermore, oral $Cl^-$ application to mice increased impulse frequencies of taste nerves connected to T1r-expressing taste cells and promoted their behavioral preferences attenuated by a T1r-specific blocker or T1r3 knock-out. These results suggest that the $Cl^-$ evokes taste sensations by binding to T1r, thereby serving as another preferred salt taste pathway at a low concentration.

**\*For correspondence:**
a_yama@okayama-u.ac.jp

†These authors contributed equally to this work

**Competing interest:** The authors declare that no competing interests exist.

## Editor's evaluation

The manuscript by Atsumi et al. presents solid evidence that identifies the T1r (sweet /umami) taste receptors as chloride ($Cl^-$) receptors. The authors employ many state-of-the-art techniques to demonstrate that T1r receptors from Medaka fish bind chloride and that this binding induces a conformational change in the heteromeric receptor. This conformational change leads to low-concentration chloride-specific action potential firing in nerves from neurons containing these receptors in mice. These results represent an important advance in our understanding of the logic of taste perception.

## Introduction

The taste sensation is initiated by specific interactions between chemicals in food and taste receptors in taste buds in the oral cavity. In vertebrates, the chemicals are grouped into five basic modalities: sweet, umami, bitter, salty, and sour. This sensation occurs through taste receptor recognition specific to a group of chemicals representing each taste modality (*Taruno et al., 2021*). Regarding the salty taste, the preferable taste, ~100 mM concentration of table salt, is evoked by specific interaction

**eLife digest** Humans perceive taste when proteins called taste receptors on the surface of the tongue are activated by molecules of food. These receptors turn on nerve cells that send signals the brain can read as sweet, sour, salty, bitter, or umami, depending on which receptor was activated. Most animals with backbones share the same five types of taste receptors.

In food, salty flavors are usually the result of adding table salt, which has two components: a sodium ion and chloride ion. The main taste receptors that signal to the brain that a food is salty become activated when they bind to the sodium ion. However, some studies have shown that salt is also perceived as sweet when eaten in minuscule amounts. It is poorly understood why this happens, but it is possible that the chloride half of salt drives the sweet taste.

In 2017, scientists worked out the structure of a taste receptor from a fish, that is equivalent to the sweet receptor in humans. Curiously, one part of this receptor, known as T1r2a/T1r3LBD, was bound to a chloride ion. This prompted Atsumi, Yasumatsu et al. to think about the 'sweet' taste of salt, leading them to take a closer look at T1r2a/T1r3LBD and whether chloride could indeed activate it.

Atsumi, Yasumatsu et al. used structural biology techniques to examine T1r2a/T1r3LBD and found evidence that the receptor might be binding chloride. Further biophysical experiments confirmed that chloride does indeed bind to the receptor, and that it also causes it to change shape. Usually, changes in shape are hallmarks of receptor activation, suggesting that chloride may activate T1r2a/T1r3LBD.

Next, Atsumi, Yasumatsu et al. checked whether chloride could stimulate the neurons that signal when food tastes sweet, by using an approach known as electrophysiology to measure the activity of these neurons in mice. The results showed that the neurons became active when a solution containing small amounts of chloride was placed on the mouse's tongue. This activity went away when a compound that can block the receptor's activity was delivered alongside the chloride. Additionally, when mice were given a choice of plain water or water containing chloride, they seemed to prefer the latter. This confirmed that mice recognized the sweetness of chloride via the activation of sweet taste receptors and neurons.

Based on these findings, Atsumi, Yasumatsu et al. propose that small amounts of salt may taste sweet because the chloride ions in the salt activate sweet taste receptors and their linked neurons. Their results also suggest that animals sense salt in many ways, likely because balanced salt levels are essential for the body to work properly. Future experiments on human taste receptors may reveal how these pathways help assess salt levels in humans.

between the epithelial sodium channel (ENaC) and sodium ion (*Chandrashekar et al., 2010*; *Figure 1*). Notably, salt sensation exhibits multifaced properties (*Roper, 2015*), thereby suggesting the existence of an adequate concentration range for salt intake to maintain the homeostasis of body fluid concentration. For example, high concentrations of salt over levels perceived as a preferred taste, such as ~500 mM, stimulate bitter and sour taste cells and are perceived as an aversive taste (*Oka et al., 2013*). Conversely, low salt concentrations under the 'preferable' concentration, such as several mM to a hundred mM concentrations, are perceived as sweet by human panels (*Bartoshuk et al., 1964*; *Bartoshuk et al., 1978*; *Cardello, 1979*). However, its mechanism has never been extensively pursued. The various impacts of the salt taste sensation indicate multiple salt detection pathways in taste buds (*Roper, 2015*). Moreover, another component of table salt, the chloride ion, participates in taste sensation because of the existence of the 'anion effect': the salty taste is most strongly perceived when the counter anion is a chloride ion (*Ye et al., 1991*). Several reports suggested a certain cellular/molecular machinery underlying anion-sensitive $Na^+$ detection or $Cl^-$ detection, which is independent of ENaC (*Lewandowski et al., 2016*; *Roebber et al., 2019*). Indeed, a recent study indicated that transmembrane channel-like 4 (TMC4) expressed in taste buds involves high-concentration $Cl^-$ sensation (*Kasahara et al., 2021*). Nevertheless, no candidate molecule capable of sensing low or preferable concentrations of $Cl^-$ has been elucidated. Therefore, the complete understanding of salt taste sensation, including the mechanism of chloride ion detection, remains unclear.

Unlike salt taste sensation, those for nutrients as sugars, amino acids, and nucleotides are understood as sweet and umami sensations through specific receptor proteins (*Li et al., 2002*; *Nelson*

| NaCl conc. | 0.1 | 1 | 10 | | 100 | | 1000 | (mM) |
|---|---|---|---|---|---|---|---|---|
| Taste sensation | | sweet | | | | | | |
| | | | salty (preferable) | | | | | |
| | | | | | | salty (aversive) | | |
| Taste cell | ? | | salty | | | sour, bitter | | |
| Taste receptor | ? | | ENaC | | | ? | | |

**Figure 1.** Salt taste sensation. Approximate concentration ranges of salt taste perceptions in humans (*Bartoshuk et al., 1978*) and qualities of taste sensation with known cells and receptors responsible for their sensing are summarized.

*et al., 2002*; *Nelson et al., 2001*). Sweet and umami receptors are composed of taste receptor type 1 (T1r) proteins in the class C G-protein-coupled receptor (GPCR) family. In humans, the T1r1/T1r3 heterodimer serves as the umami taste receptor and responds to amino acids as ʟ-glutamate and aspartate, and nucleotides. In contrast, the T1r2/T1r3 heterodimer is the sweet taste receptor and responds to sugars. We previously elucidated the crystallographic structure of the medaka fish T1r2a/T1r3 extracellular ligand-binding domain (LBD) (*Nuemket et al., 2017*), which is currently the sole reported structure of T1rs. In the structure, the amino acid binding was observed in the middle of the LBD of T1r2a and T1r3 subunits, which was consistent with the fact that T1r2a/T1r3 is an amino acid receptor (*Oike et al., 2007*). Furthermore, chloride ion binding was found in the vicinity of the amino acid-binding site in T1r3 (*Figure 2A*). So far, the physiological significance of $Cl^-$ binding for T1rs functions remains unexplored. Nevertheless, chloride ions regulate other receptors in class C GPCRs, such as metabotropic glutamate receptors (mGluRs) and calcium-sensing receptors (CaSRs), and act as positive modulators for agonist binding (*Eriksen and Thomsen, 1995*; *Kuang and Hampson, 2006*; *Liu et al., 2020*; *Tora et al., 2018*; *Tora et al., 2015*). The potential effect of $Cl^-$ binding on T1r receptor function is of significant interest under these conditions.

Here, we investigated the $Cl^-$ actions on T1rs using structural, biophysical, and physiological analyses. The $Cl^-$ binding to the LBD was investigated using the medaka fish T1r2a/T1r3LBD, which is amenable to structural and biophysical analyses. Since the $Cl^-$-binding site in T1r3 was conserved across various species, taste nerve recordings from mice were used to investigate the physiological significance of $Cl^-$. The results suggest that $Cl^-$ induces the moderate response via T1rs, thereby implying that T1rs are involved in $Cl^-$ sensation in taste buds.

## Results

### $Cl^-$-binding site in T1r3

In the previously reported structure of T1r2a/T1r3LBD crystallized in the presence of NaCl, bound $Cl^-$ was identified based on electron density and binding distances (*Nuemket et al., 2017*). To verify the $Cl^-$ binding, $Cl^-$ in the T1r2a/T1r3LBD crystal was substituted with $Br^-$, a halogen ion amenable for specific detection by anomalous scattering using a synchrotron light source (*Table 1*). The diffraction data from the crystal resulted in an anomalous difference Fourier peak at 14.1 $\sigma$ at the site in the vicinity of the amino-acid-binding site in T1r3, where $Cl^-$ was originally bound (*Figure 2B*, *Figure 2—figure supplement 1*). For further confirmation, the anomalous data of the original crystal containing $Cl^-$ were collected at 2.7 Å, where the anomalous peak for Cl and several other elements such as Ca or S can be detected (*Table 1*). The resultant anomalous difference Fourier map showed a peak at the bound $Cl^-$ position, while all the other peaks were observed at the S atoms in the protein (*Figure 2C*, *Figure 2—figure supplement 1*). These results verify that the site is able to bind halogen ions, likely accommodating $Cl^-$ under physiological conditions.

$Cl^-$ was coordinated at the binding site by the side-chain hydroxyl group of Thr105 and the main-chain amide groups of Gln148 and Ser149 (*Figure 2D*). These main-chain coordinating residues are followed by Ser150, a critical residue for binding amino acid ligands (*Nuemket et al., 2017*). Furthermore, the loop regions where Thr105 and the Gln148–Ser150 locate are followed by helices B and C,

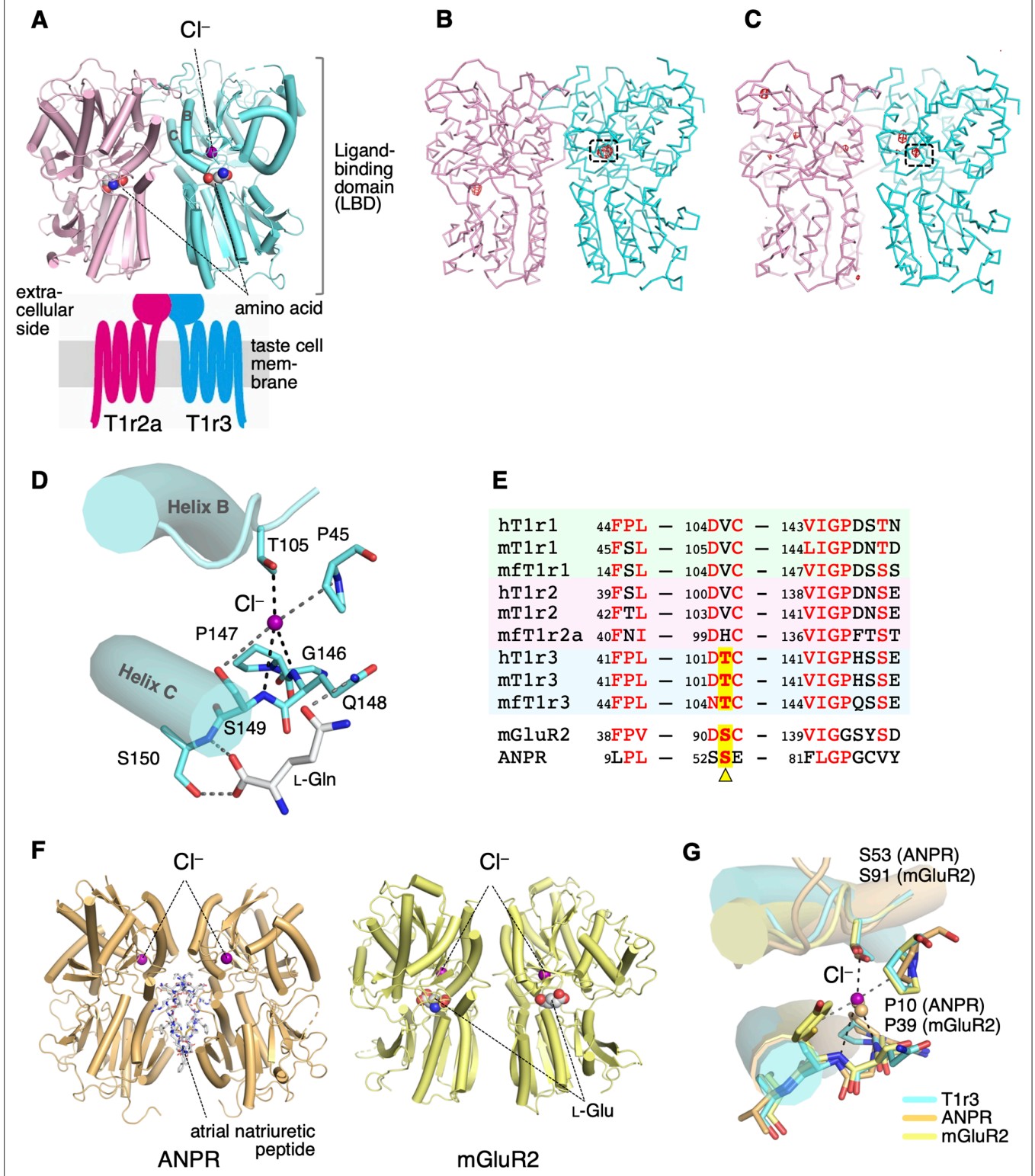

**Figure 2.** Cl⁻-binding sites in the medaka fish taste receptor T1r2a/T1r3LBD. (**A**) Schematic drawing of the overall architecture of T1r2a/T1r3. The crystal structure (PDB ID: 5X2M) (**Nuemket et al., 2017**) is shown at the ligand-binding domain (LBD) region, and helices B and C in T1r3 are labeled. (**B**) Anomalous difference Fourier map (4.5 σ, red) of the Br⁻-substituted T1r2a/T1r3LBD crystal. (**C**) Anomalous difference Fourier map (4.5 σ, red) of the Cl⁻-bound T1r2a/T1r3LBD crystal derived from the diffraction data collected at the wavelength of 2.7 Å. In panels B and C, the site originally identified the Cl⁻ binding was framed. (**D**) A close-up view of the Cl⁻-binding site in T1r3LBD in the Cl⁻-bound T1r2a/T1r3LBD (PDB ID: 5X2M). (**E**) Amino acid sequence alignment of T1r proteins and the related receptors at the Cl⁻-binding site. The 'h', 'm', and 'mf' prefixes to T1rs indicate human, mouse, and

*Figure 2 continued on next page*

*Figure 2 continued*

medaka fish, respectively. The position corresponding to Thr105 in T1r3 from medaka fish is highlighted. (**F**) The structures of atrial natriuretic peptide receptor (ANPR) (PDB ID: 1T34, left) (*Ogawa et al., 2004*) and mGluR2 (PDB ID: 5CNI, right) (*Monn et al., 2015a*) bound with Cl⁻. (**G**) Superposition of the Cl⁻-binding site in T1r3, ANPR, and mGluR2.

The online version of this article includes the following source data and figure supplement(s) for figure 2:

**Source data 1.** The anomalous difference Fourier maps shown in *Figure 2B, C*.

**Figure supplement 1.** The structure of the regions relating to the Cl⁻-binding site in medaka fish T1r2a/T1r3LBD.

respectively. These helices are essential structural units at the heterodimer interface (*Nuemket et al., 2017*; *Figure 2A*). They are known to reorient upon agonist binding, resulting in conformation rearrangement of the subunits in the dimer, likely inducing receptor activation in class C GPCRs (*Koehl et al., 2019*; *Kunishima et al., 2000*). In addition, the side-chain hydroxyl group of Ser149, which serves as a cap for the positive helix dipole of helix C, simultaneously functions as a distal ligand for Cl⁻ coordination. Therefore, the Cl⁻ binding at this site is important for organizing the structure of the amino-acid-binding site and the heterodimer interface.

The Cl⁻-binding site observed in the crystal structure is most likely conserved among T1r3s in various organisms, such as humans (*Figure 2E*). Thr105, the residue that provides the side-chain-coordinating ligand for Cl⁻ binding, is strictly conserved as either serine or threonine among T1r3s. Additionally, the amino acid sequence motifs surrounding the main-chain-coordinating ligands, FP$^{45}$L and VIGP$\zeta^{148}\zeta^{149}$, where '$\zeta$' is a hydrophilic amino acid, are well conserved to present the main-chain amide groups to coordinate Cl⁻ with an appropriate geometry. Notably, the site structurally corresponds to the Cl⁻-binding site in the hormone-binding domain of the atrial natriuretic peptide receptor (ANPR) (*Figure 2F, G*), in which Cl⁻ positively regulates the peptide hormone binding (*Misono, 2000*). Although ANPR is not a member of class C GPCR, the hormone-binding domain in ANPR shares a similar structural fold with LBD of T1rs and other class C GPCRs, and bacterial periplasmic-binding proteins (*Kunishima et al., 2000*; *van den Akker et al., 2000*). Accordingly, the conservation of the structure and the sequence motif at the Cl⁻-binding site at ANPR is also observed on mGluRs (*Ogawa et al., 2010*), and the importance of the site for receptor activation has been proposed (*Acher et al., 2011*). Indeed, Cl⁻ binding at the site corresponding to that in T1r3 was observed in several mGluR and CaSR structures (*Monn et al., 2015b*; *Zhang et al., 2016*; *Figure 2F,*

**Table 1.** X-ray data collection statistics of T1r2a/T1r3LBD–Fab16A complex.

|  | Br⁻ bound | Cl⁻ bound |
| --- | --- | --- |
| Beamline | SPring-8 BL41XU | Photon Factory BL-1A |
| Detector | PILATUS6M | EIGER X4M |
| Wavelength (Å) | 0.9194 | 2.7 |
| Space group | $P2_12_12_1$ | $P2_12_12_1$ |
| Cell dimensions |  |  |
| *a* (Å) | 102.8 | 102.8 |
| *b* (Å) | 121.6 | 120.8 |
| *c* (Å) | 129.9 | 129.1 |
| Resolution (Å) | 50–3.41 (3.43–3.41) | 49.8–3.32 (3.33–3.32) |
| $R_{sym}$ (%)* | 0.094 (0.808) | 0.089 (0.865) |
| $I/\sigma(I)$* | 16.4 (2.3) | 15.1 (2.5) |
| Completeness (%)* | 99.8 (99.2) | 99.6 (97.8) |
| Redundancy* | 7.0 (7.0) | 6.9 (6.8) |

*Values in parentheses refer to data in the highest resolution shells.

*G*) and was identified as a site regulating agonist binding (*Liu et al., 2020*; *Tora et al., 2015*). These results strongly imply the possibility that Cl⁻ has some actions on T1r receptor functions.

In contrast, conservation at the Thr105 position was not observed among T1r1 and T1r2 (*Figure 2E*). Evidently, no significant anomalous peak derived from Br⁻ or Cl⁻ binding was observed in the crystal structure at the corresponding site in T1r2a (*Figure 2B, C*). His100 in T1r2a, which corresponds to Thr105 in T1r3, adopted a significantly different side-chain conformation from that of Thr105 in T1r3 (*Figure 2—figure supplement 1*). Therefore, T1r1 and T1r2's ability to bind Cl⁻ is unlikely.

In addition to the Cl⁻-binding site discussed above, the Br⁻-substituted crystal exhibited an anomalous peak at 8.5 $\sigma$ in T1r2a, at a position close to the Lys265 side-chain $\varepsilon$-amino group (*Figure 2B*, *Figure 2—figure supplement 1*). Nevertheless, Cl⁻ binding was not observed in the original Cl⁻-contained crystal. This is further confirmed by the absence of an anomalous peak at this position in the data collected at 2.7 Å (*Figure 2C*, *Figure 2—figure supplement 1*). Therefore, the site might have the ability to bind anions such as Br⁻ or larger; but be not specific to Cl⁻. In human T1r1, the residue corresponding to Lys265 (Arg277) was suggested as a critical residue for activities of inosine monophosphate, an umami enhancer (*Zhang et al., 2008*).

## Cl⁻-binding properties in T1r2a/T1r3LBD

To investigate the Cl⁻ actions on T1r functions, we first examined the properties of the Cl⁻ binding to medaka T1r2a/T1r3LBD using various biophysical techniques. For this purpose, the purified T1r2a/T1r3LBD was subjected to differential scanning fluorimetry (DSF), which we previously used for the amino acid-binding analysis (*Yoshida et al., 2019*). In order to prepare a Cl⁻-free condition, Cl⁻ in the sample was substituted with gluconate, as it is unlikely accommodated in the Cl⁻ site due to its much larger size. We confirmed that gluconate does not serve as a ligand for T1r2a/T1r3LBD (*Figure 3—figure supplement 1*).

The addition of Cl⁻ to the Cl⁻-free T1r2a/T1r3LBD sample resulted in thermal stabilization of the protein (*Figure 3A*), which is indicative of Cl⁻ binding to the protein. The apparent $K_d$ value for Cl⁻ estimated by the melting temperatures ($T_m$) at various Cl⁻ concentrations was ~110 μM (*Figure 3B*, *Table 2*). The Cl⁻-dependent thermal stabilization was confirmed by the fluorescence-detection size-exclusion chromatography-based thermostability (FSEC-TS) assay (*Hattori et al., 2012*; *Figure 3C*, *Table 2*). However, the Cl⁻-dependent stabilization was not observed on T1r2a/T1r3 with the Cl⁻-site mutation, T105A in T1r3. In the case of this mutant, the $T_m$ values for both in the presence and absence of Cl⁻ was similar to the values obtained for the wild-type (WT) protein in the absence of Cl⁻ (*Figure 3C*, *Figure 3—figure supplement 1*, *Table 2*). These results indicate that the Cl⁻ effect attributed to the identical site where the Cl⁻ binding was observed in the crystal structure of T1r3.

Next, we examined the consequence of the Cl⁻ binding to T1r2a/T1r3LBD via Förster resonance energy transfer (FRET) using the fluorescent protein-fused sample. Class C GPCRs commonly exhibit agonist-induced conformational changes in LBD, such as the dimer rearrangement, which is essential for receptor activation and signaling (*Ellaithy et al., 2020*; *Koehl et al., 2019*; *Kunishima et al., 2000*; *Lin et al., 2021*). Consistent with this, we previously reported that T1r2a/T1r3LBD shows conformational change concomitant with the binding of amino acids, which can be detected as increased FRET intensity (*Nango et al., 2016*). Notably, adding Cl⁻ to the fluorescent protein-fused T1r2a/T1r3LBD also increased FRET intensities, similar to amino acids (*Figure 3D*). The EC₅₀ for Cl⁻-induced FRET signal change was determined as ~1 mM (*Table 2*). Note that both DSF and FRET estimations have some degree of error: the former produced slightly lower values than the latter, particularly in the case of weak affinities in the mM concentration range (*Yoshida et al., 2019*). As such, although the EC₅₀ value determined by FRET was slightly higher than the apparent $K_d$ value of Cl⁻ determined by DSF, the two are most likely relevant. Considering that the Cl⁻-binding site is located adjacent to the dimer interface, which exhibits reorientation upon agonist binding (*Figure 2D*), the results suggest that the Cl⁻ binding to T1r2a/T1r3LBD induces a conformational rearrangement of T1r2a/T1r3LBD similar to its agonist amino acid. Nevertheless, the extent of the FRET change induced by Cl⁻ was smaller than the changes induced by amino acids, such as ~1/2 of the latter (*Figure 3E*). Therefore, the results suggest that the extent of the conformational change induced by Cl⁻ is smaller than the change induced by amino acids. The extent of Cl⁻-dependent FRET index change was reduced on T1r2a/T1r3 with the Cl⁻-site mutation, T105A in T1r3 (*Figure 3E*). Considering that the amino-acid-dependent change in the mutant was also significantly reduced (*Figure 3E*), the T105A mutation on

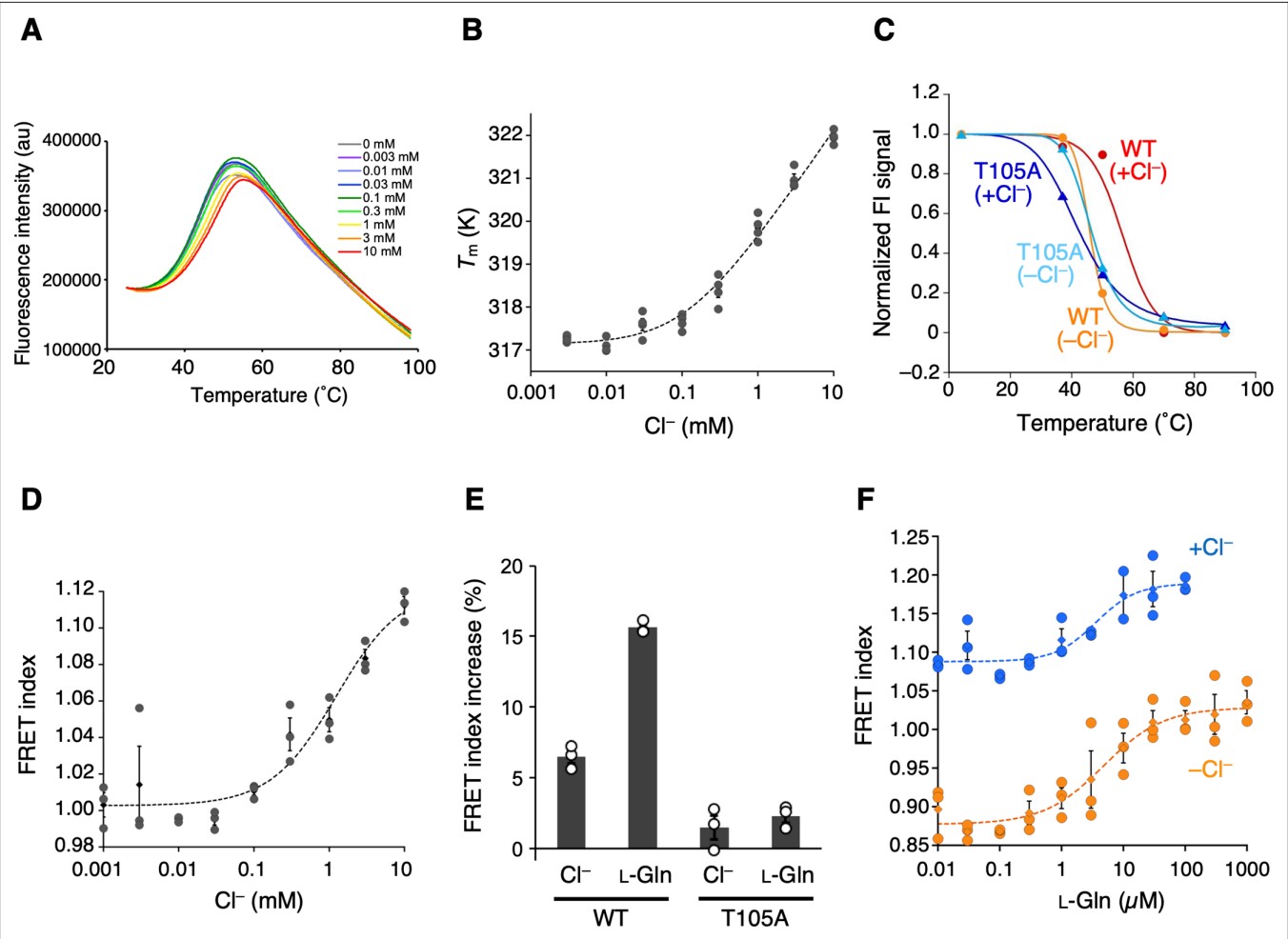

**Figure 3.** The Cl⁻-binding properties of T1r2a/T1r3LBD. (**A**) Representative thermal melt curves of T1r2a/T1r3LBD in the presence of 0.003–10 mM concentrations of Cl⁻ measured using differential scanning fluorimetry (DSF). (**B**) Dose-dependent $T_m$ changes of T1r2a/T1r3LBD by addition of Cl⁻ ($n$ = 4). (**C**) Thermal melting curves of wild-type (WT) and the T1r3-T105A mutant of T1r2a/T1r3LBD in the presence and absence of Cl⁻, analyzed by fluorescence-detection size-exclusion chromatography-based thermostability (FSEC-TS) assay ($n$ = 1). (**D**) Dose-dependent Förster resonance energy transfer (FRET) signal changes of the T1r2aLBD-Cerulean and T1r3LBD-Venus heterodimer by addition of Cl⁻ ($n$ = 3). (**E**) FRET index increases by adding 10 mM Cl⁻ or 1 mM ʟ-glutamine to the WT or T1r3-T105A mutant T1r2aLBD-Cerulean/T1r3LBD-Venus heterodimer relative to that in the absence of any ligand in the absence of Cl⁻ ($n$ = 3). (**F**) Dose-dependent FRET signal changes of the T1r2aLBD-Cerulean and T1r3LBD-Venus heterodimer induced by the addition of ʟ-glutamine in the presence and absence of Cl⁻ ($n$ = 3). The experiments were performed two (panels A, B, D, E), three (C and WT), four (F and +Cl⁻ condition), or one (C and mutant; F and −Cl⁻ condition) time(s), and the results from one representative experiment are shown with numbers of technical replicates. Data points represent mean (panels B, D, F: diamonds; panel E: bars) ± standard error of the mean (SEM).

The online version of this article includes the following source data and figure supplement(s) for figure 3:

**Source data 1.** Excel file with numerical data used for *Figure 3*.

**Figure supplement 1.** The properties of T1r2a/T1r3LBD in the presence and absence of Cl⁻.

---

T1r3 might result in losing the ability of the conformational change induced by Cl⁻ and amino acid bindings, although the possibility of deactivation of the protein during preparation due to its low stability cannot be excluded.

In addition to the Cl⁻-binding effect, the Cl⁻ effect on amino acid binding to T1r2a/T1r3LBD was investigated by FRET and isothermal calorimetry (ITC). The $K_d$ values for ʟ-glutamine binding determined by ITC, as well as the $EC_{50}$ values and the other parameters for ʟ-glutamine-induced conformational change determined by FRET, did not differ in the presence and absence of Cl⁻ (*Figure 3F*, *Figure 3—figure supplement 1*, and *Table 2*). These results indicated that the Cl⁻ binding had no significant effect on the binding of ʟ-glutamine, a representative taste substance, at least for T1r2a/T1r3LBD from medaka fish.

**Table 2.** Properties of the Cl$^-$ binding to T1r2a/T1r3LBD.

| Cl$^-$ binding, DSF ($n = 4$) | | |
|---|---|---|
| $K_{\text{d-app}}$ (mM) | 0.111 ± 0.046 | |
| **Protein thermal stability, DSF ($n = 6$)** | | |
| Condition | +Cl$^-$ | −Cl$^-$ |
| Melting temperature (°C) | 55.2 ± 0.03 | 46.6 ± 0.06 |
| **Protein thermal stability, FSEC-TS** | | |
| Wild-type T1r2a/T1r3LBD | | |
| Condition | +Cl$^-$ | −Cl$^-$ |
| Melting temperature (°C) | 56.4 ± 5.1 | 46.0 ± 0.3 |
| Mutant T1r2a/T1r3-T105ALBD | | |
| Condition | +Cl$^-$ | −Cl$^-$ |
| Melting temperature (°C) | 42.7 ± 0.1 | 46.7 ± 0.7 |
| **Cl$^-$ binding, FRET ($n = 3$)** | | |
| FRET index minimum | 1.00 ± 0.005 | |
| FRET index change | 0.119 ± 0.014 | |
| EC$_{50}$ (mM)* | 1.23 ± 0.53 | |
| **L-Glutamine binding, FRET ($n = 3$)** | | |
| Condition | +Cl$^-$ | −Cl$^-$ |
| FRET index minimum | 1.09 ± 0.01 | 0.88 ± 0.01 |
| FRET index change | 0.10 ± 0.02 | 0.15 ± 0.02 |
| EC$_{50}$ (µM) | 3.59 ± 1.74 | 4.78 ± 1.41 |
| Hill coefficient | 1.31 ± 0.74 | 0.90 ± 0.22 |
| **L-Glutamine binding, ITC** | | |
| Condition | +Cl$^-$ | −Cl$^-$ |
| $N$ (sites) | 0.389 ± 0.028 | 0.303 ± 0.023 |
| $K_{\text{a}}$ (M$^{-1}$) [converted to $K_{\text{d}}$ (µM)] | (2.85 ± 0.65) × 10$^5$ [3.51] | (2.12 ± 0.46) × 10$^5$ [4.72] |
| $\Delta H$ (kcal/mol) | −12.3 ± 1.2 | −12.9 ± 1.3 |
| $\Delta S$ (cal/mol/deg) | −16.3 | −18.8 |

*Hill coefficient was fixed to 1 for fitting.

## Taste response to Cl$^-$ through T1rs in mouse

Biophysical studies on T1r2a/T1r3LBD from medaka fish suggested that Cl$^-$ binding to T1r3LBD induces a conformational change similar to that of an agonist without affecting agonist binding. As described above, the Cl$^-$-binding site is likely conserved among T1r3 in various species, such as those in mammals. Therefore, we analyzed single fiber responses from mouse chorda tympani (CT) nerve to investigate the physiological effect of Cl$^-$ on taste sensation. While conventional cell-based receptor assay systems are affected by changes in extracellular ionic components, the application of various solutions to the taste pore side of the taste buds projected to taste nerve systems, are transduced exclusively as 'taste' signals, without inducing the other cellular responses derived from the ionic component changes in the surrounding environment.

We first identified nerve fibers that connect to T1r-expressing taste cells in WT mice, that is, that receive taste information from cells likely possessing sweet (T1r2/T1r3) and umami (T1r1/T1r3) receptors (*Yasumatsu et al., 2012*). The identification was evidenced by responses to T1r agonists, such as

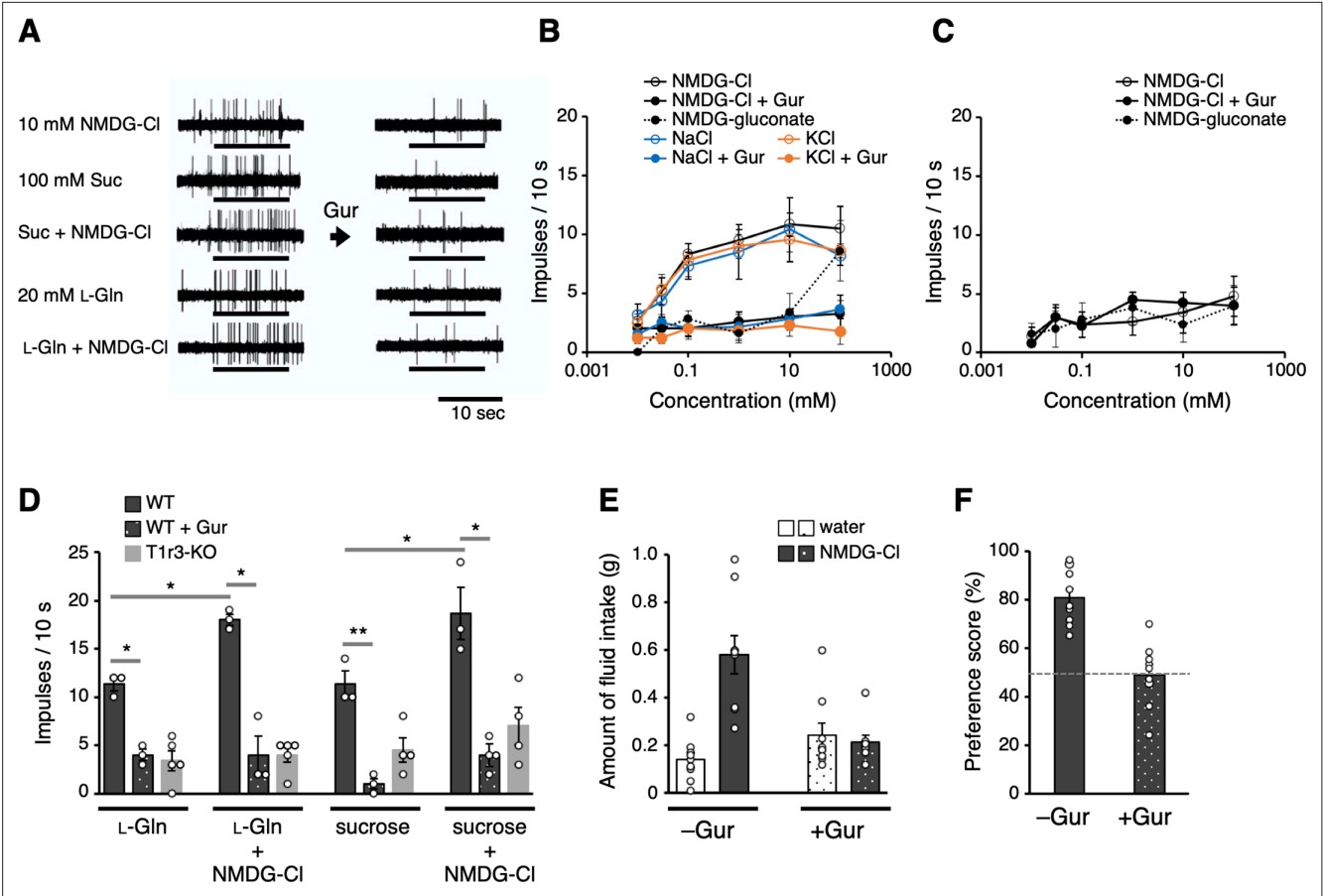

**Figure 4.** Electrophysiological and behavioral analyses of the T1r-mediated Cl⁻ responses in mouse. (**A–D**) Results of single fiber recordings from the mouse chorda tympani nerve. (**A**) Representative recordings of single fibers that connect to T1r-expressing taste cells. The stimuli were 10 mM NMDG-Cl, 100 mM sucrose, 100 mM sucrose + 10 mM NMDG-Cl, 20 mM L-glutamine, or 20 mM L-glutamine + 10 mM NMDG-Cl. Lines indicate the application of stimuli to the tongue. All the responses were suppressed by lingual treatment with a T1r blocker, Gur (right). (**B**) Impulse frequencies in response to the concentration series of NMDG-Cl, NaCl, or KCl before and after Gur treatment in wild-type (WT) mice. Responses to NMDG-gluconate are also shown. The mean number of net impulses per 10 s (mean response) ± standard error of the mean (SEM) in Gur-sensitive fibers (n = 5–6 from six mice). (**C**) Impulse frequencies in response to the concentration series of NMDG-Cl before and after Gur treatment were measured in T1r3-KO mice (n = 4–5 from three mice). Responses to NMDG-gluconate are also shown. (**D**) Impulse frequencies to 20 mM L-glutamine or 100 mM sucrose in the absence or presence of 10 mM NMDG-Cl before and after Gur treatment. Responses to 20 mM L-glutamine or 100 mM sucrose by T1r3-KO mouse are also shown. Values are mean (bars) ± SEM (n = 3–5 from three mice each). *, **: paired t-test; *p < 0.05 and **p < 0.01. (**E**) Amount of fluid intake for water and 10 mM NMDG-Cl in the two-bottle preference tests. Values are mean (bars) ± SEM (n = 9 mice). (**F**) NMDG-Cl intake shown in (**E**) normalized to water intake (preference score) in the two-bottle preference tests. A score >50% indicates that the taste solution was preferred over water.

The online version of this article includes the following source data for figure 4:

**Source data 1.** Excel file with numerical data used for **Figure 4**.

sugars and amino acids, which were inhibited by gurmarin (Gur), a T1r-specific blocker (**Daly et al., 2013**; **Margolskee et al., 2007**; **Ninomiya and Imoto, 1995**; **Ninomiya et al., 1999**). Then, we examined the responses to Cl⁻ in these fibers. Remarkably, the fibers also exhibited responses induced by Cl⁻, which was applied as a form of NMDG-Cl devoid of the known salty taste stimulant, sodium ion (**Figure 4A**). Cl⁻-induced impulse frequencies from the nerves increased in a concentration-dependent manner (**Figure 4B**). The responses to NMDG-Cl, NaCl, and KCl at the same concentrations did not differ significantly (repeated measures analysis of variance [ANOVA], p > 0.05), whereas responses to NMDG-gluconate were significantly smaller than those to NMDG-Cl (repeated measures ANOVA: $F_{(1, 43)}$ = 31.33, p < 0.001) and did not induce explicit responses up to 10 mM. These results confirmed that the observed responses were attributed to Cl⁻. All responses to Cl⁻, regardless of the type of counter cations, were significantly decreased by lingual treatment with Gur (**Figure 4B**, repeated measures ANOVA: $F_{(1, 42)}$ = 56.65, p < 0.001 for NMDG-Cl; $F_{(1, 50)}$ = 24.78, p < 0.001 for NaCl; and $F_{(1, 45)}$ = 35.72,

$p < 0.001$ for KCl). Furthermore, responses to $Cl^-$ in T1r3-KO mice were significantly lower than those in WT mice (repeated measures ANOVA: $F_{(1, 43)} = 25.36$, $p < 0.001$; **Figure 4C**). The results indicate that the observed $Cl^-$-dependent responses were mediated by T1r. Notably, the $Cl^-$-concentration range that induced nerve responses was lower ($\leq \sim 10$ mM) than that for $Na^+$ detection by ENaC when applied as the NaCl form ($\geq \sim 30$ mM) (**Chandrashekar et al., 2010**) but is consistent with those for $Cl^-$-binding and $Cl^-$-induced conformational change observed on T1r2a/T1r3LBD (**Table 2**). These results suggest that a low $Cl^-$ concentration is sensorily detected via T1r in taste buds. Although the responses induced by known taste substances for T1rs, such as sugars and amino acids, range from tens to hundreds of impulse frequencies per 10 s (**Yasumatsu et al., 2012**); the maximum response level induced by $Cl^-$ was low, $\sim 10$ per 10 s (**Figure 4B**). According to our observations, $Cl^-$ likely produces a 'light' taste sensation compared with other known taste substances.

Next, to examine the physiological interaction between a canonical taste substance for mouse T1r and $Cl^-$, we recorded responses to 20 mM L-glutamine or 100 mM sucrose from T1r1/T1r3- and T1r2/T1r3-expressing cells, respectively, with or without NMDG-Cl from the same T1r-connecting single fibers (**Figure 4A**). The concentrations for the taste substances were set to induce responses greater than the baseline but less than maximum. As shown in **Figure 4D**, the response to L-glutamine or sucrose increased significantly by adding 10 mM NMDG-Cl (paired *t*-test, $t2 = 7.56$, $p = 0.017$ for L-glutamine and $t2 = 5.05$, $p = 0.037$ for sucrose). We confirmed that these responses had been suppressed by a lingual treatment of Gur in the presence ($t2 = 6.73$, $p = 0.021$ for L-glutamine and $t2 = 8.80$, $p = 0.013$ for sucrose) or absence ($t2 = 8.32$, $p = 0.014$ for L-glutamine and $t2 = 11.72$, $p = 0.007$ for sucrose) of NMDG-Cl to a similar extent and a similar level to the responses in T1r3-KO mice. Moreover, the responses to the mixtures did not differ significantly from the summation of the responses to each solution ($t2 = 2.34$, $p = 0.145$ for L-glutamine and $t2 = 2.31$, $p = 0.147$ for sucrose). The results suggest that the simultaneous binding of $Cl^-$ and a canonical taste substance, such as amino acids and sugars, to T1r do not cause synergistic responses.

Finally, we addressed whether T1r-mediated $Cl^-$ responses observed in the taste nerves involve taste perception. Thus far, most reported behavioral assays for examining the gustatory detection of NaCl were performed with concentrations above $\sim 30$ mM, which can induce ENaC-mediated responses. Nevertheless, a few reports have shown that NaCl solution induced higher consumption relative to water even below 10 mM concentration (**Dyr et al., 2014**; **Stewart et al., 1994**), which is below the range inducing ENaC-mediated responses (**Chandrashekar et al., 2010**) but within what we observed in the $Cl^-$-induced taste nerve responses via T1r. For verification, we performed a mouse two-bottle choice test using NMDG-Cl solution and analyzed the preference relative to water. The mouse preferred water containing 10 mM NMDG-Cl, which was abolished by the application of Gur (**Figure 4E, F**). These results suggest that $Cl^-$ is preferably perceived through taste signal transduction mediated by T1r.

## Discussion

In this study, $Cl^-$ is found to specifically interact with the LBD in T1r3, a common component of sweet and umami taste receptors, and induces a conformational change in the receptor's LBD region. The $Cl^-$-induced conformational change is similar to that induced by canonical taste substances for T1rs, amino acids, though its efficacy is slightly lower. Therefore, as with other class C GPCRs, the structural change at LBD caused by $Cl^-$ binding most likely provokes receptor activation, resulting in G-protein activation in taste cells. The signal is considered to be transmitted further through the common downstream cascade of T1rs: activation of phospholipase $C\beta_2$ and resultant inositol triphosphate ($IP_3$) production, $IP_3$-dependent activation of $IP_3$ receptors followed by $Ca^{2+}$ release from endoplasmic reticulum, $Ca^{2+}$-dependent activation of TRPM5 channel inducing taste cell depolarization, and subsequent action potential generation and neurotransmitter ATP release through CALHM1/3 (**Taruno et al., 2021**). The $Cl^-$ binding to T1r in taste cells is thus probably transmitted to the sweet taste nervous system, resulting in a light yet preferable taste sensation. Since T1rs are conserved across vertebrates and the $Cl^-$-binding site is likely conserved among T1r3 in various organisms, T1r-mediated $Cl^-$ responses might be common in many animals, such as humans. Evidently, the concentration range for the $Cl^-$-induced conformational change of medaka T1r2a/T1r3LBD and increase in murine sweet nerve impulses observed in this study (i.e., $\leq \sim 10$ mM) agrees with the NaCl concentration perceived as 'sweet' by humans (**Bartoshuk et al., 1964**; **Bartoshuk et al., 1978**; **Cardello, 1979**;

*Figure 1*). Additionally, the sweet sensation induced by NaCl was reportedly suppressed by topical application of *Gymnema sylvestre* (*Bartoshuk et al., 1978*), containing Gymnemic acids, which are specific inhibitors of human sweet taste receptor T1r2/T1r3 (*Sanematsu et al., 2014*). Overall, these results agree with the involvement of T1rs in the Cl⁻-taste detection. These findings agree with an earlier hypothesis by Bartoshuk et al. that dilute NaCl contains a sweet stimulus that interacts with the same receptor molecules as sucrose (*Bartoshuk et al., 1978*). The reported 'sweet' sensation by low NaCl concentration faded out at the concentrations detected as 'salty' (*Bartoshuk et al., 1964*; *Bartoshuk et al., 1978*; *Cardello, 1979*), thereby agreeing with the fact that we might be unaware that table salt is sweet at such concentrations. The phenomenon could be explained by 'mixture suppression' (*Bartoshuk, 1975*; *Keast and Breslin, 2003*; *Stevens, 1996*), such that a light sweet sensation is masked by an intense salty sensation in a higher concentration range of NaCl.

The Cl⁻ perception at low salt concentrations found in this study is achieved via the T1rs-mediating taste system, which transduces information as a preferred taste by nature. Since Cl⁻ is also a component of table salt, this system might serve as another pathway for preferred salt perception promoting intake along with the pathway for Na⁺ perception as a preferred taste via taste cells that express the Na⁺ receptor ENaC and a puringeric neurotransmission channel CALHM1/3 (*Nomura et al., 2020*). In contrast, the T1r-mediated Cl⁻ sensing observed in this study shows several different properties from those of cells or the molecules reportedly exhibiting anion-sensitive Na⁺ detection or Cl⁻ detection in taste buds thus far. Specifically, the anion-sensitive Na⁺-responding taste cells reported by Lewandowski et al. are type 3 cells, which include sour- but not sweet-responding cells (*Lewandowski et al., 2016*). The Cl⁻-detection pathway reported by Roebber et al. was observed in type 2 cells, which include both sweet- and bitter-responding cells, but was inhibited by a blocker of phospholipase C, which mediates sweet- and bitter-signaling downstream of T1r and bitter receptors (*Roebber et al., 2019*). The Cl⁻ responses mediated by TMC4 were observed in the glossopharyngeal nerve but not in the CT nerve wherein T1r-mediated responses were observed (*Kasahara et al., 2021*). Notably, responses by these reported pathways were tested by much higher concentrations of NaCl, typically in the several hundred mM range (*Kasahara et al., 2021*; *Lewandowski et al., 2016*; *Roebber et al., 2019*), than those used in this study. These results imply that there may be multiple distinct concentration-dependent pathways for Cl⁻ detection in taste buds. Given that high NaCl concentration is transduced as an aversive taste through bitter and sour taste cells (*Oka et al., 2013*), the relevance between the reported high Cl⁻-responsive pathways and high NaCl responses by bitter and sour cells is of interest, as pointed out by the previous studies, and will require further investigation.

Salt taste sensation and natriuresis are critical physiological processes that regulate sodium intake and excretion to maintain body fluid homeostasis. Intriguingly, both processes were found to use the counter anion Cl⁻ to regulate the molecular functions of the receptors, T1rs and ANPRs, which share a similar extracellular protein architecture with a conserved Cl⁻-binding site. In the case of ANPR, mGluRs, and CaSR, positive allosteric modulations by Cl⁻ for agonist binding have been observed, with some variations in the extent of the enhancement (*Eriksen and Thomsen, 1995*; *Kuang and Hampson, 2006*; *Liu et al., 2020*; *Tora et al., 2018*; *Tora et al., 2015*). Whether the Cl⁻ actions on T1rs of other subtypes or from other organisms have some variance is yet to be examined.

## Materials and methods

### Key resources table

| Reagent type (species) or resource | Designation | Source or reference | Identifiers | Additional information |
|---|---|---|---|---|
| Strain, strain background (*Mus musculus*, male) | C57BL/6JCrj | Charles River Japan | | |
| Genetic reagent (*Mus musculus*) | T1r3GFP-KO | This study | | See Materials and methods, 'Single fiber recording from mouse chorda tympani (CT) nerve' subsection |
| Cell line (*Drosophila melanogaster*) | S2 | Invitrogen | Cat # R69007 | |
| Cell line (*Drosophila melanogaster*) | S2, high-expression clone for medaka T1r2a/T1r3LBD | DOI:10.1038/ncomms15530 | | |

*Continued on next page*

*Continued*

| Reagent type (species) or resource | Designation | Source or reference | Identifiers | Additional information |
|---|---|---|---|---|
| Cell line (*Drosophila melanogaster*) | S2, high-expression clone for medaka T1r2aLBD-Cerulean/T1r3LBD-Venus | DOI:10.1038/ncomms15530 | | |
| Antibody | Anti-medaka T1r2a, clone 16 A (mouse monoclonal) | DOI:10.1038/ncomms15530 | | Fab fragment was used for crystallization. See Materials and methods, 'Crystallography' subsection |
| Recombinant DNA reagent | pAc-mfT1r3L-Ve | DOI:10.1038/srep25745 | | |
| Recombinant DNA reagent | pAc-mfT1r2aL-Ce | DOI:10.1038/srep25745 | | |
| Recombinant DNA reagent | pAc-mfT1r3L | DOI:10.1002/pro.3271 | | |
| Recombinant DNA reagent | pAc-mfT1r2aL | DOI:10.1002/pro.3271 | | |
| Sequence-based reagent | PCR primer used for T1r3-T105A mutation | This paper | | TAC AAC GCG TGC AGA CAC TCA GCT GTT ATT G |
| Sequence-based reagent | PCR primer used for T1r3-T105A mutation | This paper | | TCT GCA CGC GTT GTA GAT TTT ATA ACC CAA C |
| Peptide, recombinant protein | FLAG peptide | PH Japan | peptide | DYKDDDDK |
| Peptide, recombinant protein | gurmarin | DOI:10.1016/0300-9629(91)90,475r | peptide | Prof. Yuzo Ninomiya |
| Commercial assay or kit | Protein Thermal Shift Dye Kit | Thermo Fisher | Cat # 4461146 | |
| Software, algorithm | XDS | DOI:10.1107/S0907444909047337 | | |
| Software, algorithm | PHASER | DOI:10.1107/S0021889807021206 | | |
| Software, algorithm | Protein Thermal Shift Software | Applied Biosystems | Version 1.3 | |
| Software, algorithm | KaleidaGraph | Synergy Software | | |
| Software, algorithm | ORIGIN | OriginLab | | |

## Cell lines

All protein samples used for structural and functional analyses were prepared using *Drosophila* S2 cells (Invitrogen) or high-expression clones for each protein sample established from S2 cells in previous studies (*Nuemket et al., 2017*; *Yamashita et al., 2017*). No authentication or test for mycoplasma contamination was performed. We verified the cell line identities and conditions by checking the expression levels and characteristics of the recombinant proteins of interest.

## Crystallography

The L-glutamine-bound medaka T1r2a/T1r3LBD crystals, in complex with a crystallization chaperone Fab16A, were prepared in the presence of NaCl as described (*Nuemket et al., 2017*). For the preparation of the Br⁻-substituted crystals, the obtained crystals were soaked in a mother liquor consisting of 100 mM MES–NaOH, pH 6.0, 50 mM NaBr, 17% PEG1500, 5% PEG400, 5 mM L-glutamine, 2 mM calcium acetate, cryoprotected by gradually increasing the concentration of glycerol to 10%, incubated for 2 hr, and flash-frozen.

The X-ray diffraction data were collected at the SPring-8 beamline BL41XU using a PILATUS6M detector (DECTRIS) at wavelength 0.9194 Å or at the Photon Factory beamline BL-1A using an EIGER X4M detector (DECTRIS) at wavelength 2.7 Å. The data were processed with XDS (*Kabsch, 2010*; *Table 1*). The phases for anomalous difference Fourier map calculation were obtained by molecular replacement methods with the program PHASER (*McCoy et al., 2007*), using the structures of a single

unit of the T1r2a/T1r3LBD-Fab16A complex (PDB ID: 5X2M; ligands and water models were removed) (*Nuemket et al., 2017*) as the search model (the $R/R_{free}$ of the model for the Br$^-$ data collected at 0.9194 Å: 0.248/0.350; the Cl$^-$ data collected at 2.7 Å: 0.233/0.339).

## Differential scanning fluorimetry

DSF was performed as previously described (*Yoshida et al., 2019*). The purified medaka T1R2a/3LBD heterodimer protein was prepared (*Nango et al., 2016*) and dialyzed with buffer A (20 mM HEPES–NaOH, 300 mM sodium gluconate, pH 7.5) to remove Cl$^-$. 1 µg of the dialyzed protein sample was mixed with Protein Thermal Shift Dye (Applied Biosystems) and 0.003–10 mM NaCl in 20 µl of buffer A. The mixture solutions were then loaded to a MicroAmp Fast Optical 48-Well Reaction Plate (Applied Biosystems). Fluorescent intensities were measured by the StepOne Real-Time PCR System (Applied Biosystems) while the temperature raised from 25 to 99°C with a velocity of 0.022°C/s. For detection, the reporter and quencher were set as 'ROX' and 'none', respectively. The apparent melting transition temperature ($T_m$) was determined using the maximum of the derivatives of the melt curve (dFluorescence/d$T$) by Protein Thermal Shift Software version 1.3 (Applied Biosystems). The apparent dissociation constant ($K_{d-app}$) for Cl$^-$ derived from the $T_m$ values at different NaCl concentrations was estimated using a thermodynamic model proposed by *Schellman, 1975* as described (*Yoshida et al., 2019*). The sample sizes for the analyses by DSF, as well as FRET and isothermal titration calorimetry described below, were set to obtain reliable values based on the experiences in the previous studies (*Nango et al., 2016*; *Nuemket et al., 2017*; *Yoshida et al., 2019*).

## Förster resonance energy transfer analysis

FRET analysis was performed as described previously (*Nango et al., 2016*; *Nuemket et al., 2017*). The WT medaka T1r2aLBD-Cerulean and T1r3LBD-Venus fusion heterodimer proteins were prepared as previously described (*Nuemket et al., 2017*). While the yellow fluorescent protein, a commonly used FRET acceptor, is sensitive to halides (*Wachter and Remington, 1999*), Venus is a halide-insensitive variant with the $K_d$ value for Cl$^-$ as >10$^4$ mM (*Nagai et al., 2002*). Therefore, fluorescence changes due to chloride ions themselves, which are not due to energy transfer of the fluorescent proteins, are most likely negligible under the conditions tested in this study. For the mutant protein preparation, a T1r3-T105A mutation was introduced into the vector pAc-mfT1r3aL-Ve (*Nango et al., 2016*) using polymerase chain reaction. The mutant expression vector was co-introduced with pAc-mfT1r2aL-Ce (*Nango et al., 2016*) to *Drosophila* S2 cells using polyethyleneimine (PEI) 'MAX' (Polysciences) as previously described (*Bleckmann et al., 2019*) with a ratio of 0.5 µg pAc-mfT1r2aL-Ce, 0.5 µg pAc- mfT1r3aL-Ve-T105A, and 10 µg PEI to ~1 × 10$^6$ cells. Protein expression and purification were conducted similarly as for the WT protein.

For the Cl$^-$ titration, the purified protein samples were dialyzed against buffer A in the presence of 1 mM L-alanine. Afterward, the samples were diluted with buffer A to reduce the remaining L-alanine concentration below 1 µM (<~1/100 of EC$_{50}$; *Nango et al., 2016*; *Nuemket et al., 2017*), and then incubated in the presence of 0.001–10 mM NaCl or 1 mM L-glutamine in buffer A at 4°C overnight. For L-glutamine titration in the presence or absence of Cl$^-$, the protein solution was dialyzed with buffer B (20 mM HEPES–Tris, 300 mM NaCl, pH 7.5) or buffer C (20 mM HEPES–Tris, 300 mM sodium gluconate, pH 7.5) in the presence of 1 mM L-alanine to prepare the conditions with or without Cl$^-$, respectively. Then, the samples were diluted with buffer B or C to reduce the remaining L-alanine concentration below 1 µM and then incubated in the presence of 0.01–1000 µM L-glutamine at 4°C overnight. Fluorescence spectra were recorded at 298 K with a FluoroMax4 spectrofluorometer (Horiba). The sample was excited at 433 nm, and FRET was detected via the emission at 526 nm. The emission at 475 nm was also recorded for the FRET index calculation. The FRET index (intensity at 526 nm/intensity at 475 nm) was plotted against the Cl$^-$ or L-glutamine concentration, and the titration curves were fitted to the Hill equation using KaleidaGraph (Synergy Software).

## Isothermal titration calorimetry

In order to prepare the conditions with or without Cl$^-$, the purified medaka T1R2a/3LBD heterodimer protein was dialyzed with buffer B or C, respectively. The dialyzed protein solution (~50 µM) was then loaded into the sample cell in iTC200 (GE Healthcare) after the removal of insoluble materials by centrifugation (10,000 × $g$, 15 min, 277 K). The titration was performed by injecting 2 µl of 400 µM

L-glutamine at intervals of 120 s at 298 K. The thermograms and the binding isotherms were analyzed with Origin software (OriginLab), assuming one set of binding sites for fitting.

## Fluorescence-detection size-exclusion chromatography-based thermostability assay

A T1r3-T105A mutation was introduced in the vector pAc_mfT1r3L (*Yamashita et al., 2017*) by PCR. The mutant expression vector was co-introduced with pAc_mft1r2aL (*Yamashita et al., 2017*) to *Drosophila* S2 cells to establish a stable high-expression clone cell as previously described (*Yamashita et al., 2017*). The WT T1r2a/T1r3LBD and mutant T1r2a/T1r3-105A-LBD proteins were expressed and purified as previously described (*Nango et al., 2016*) with several modifications as listed below. After the protein binding to ANTI-FLAG M2 affinity gel (SIGMA), the resin was washed with either buffer D (20 mM HEPES–NaOH, 0.3 M NaCl, 2 mM CaCl$_2$, 5 mM L-Gln, pH 7.5) or buffer E (20 mM HEPES–NaOH, 0.3 M Na gluconate, 2 mM Ca gluconate, 5 mM L-Gln, pH 7.5). Then, the protein was eluted with 100 µg/ml FLAG peptide in buffer D or E.

The protein solutions (50 µg/ml) in buffer D or E were incubated at 4, 37, 50, 70, or 90°C at 2 hr. Subsequently, the samples were loaded on an SEC-5 column, 500 Å, 4.6 × 300 mm (Agilent) connected to a Prominence HPLC system (Shimadzu), using buffer D or E as a running buffer at a flow rate of 0.3 ml/min. The elution profiles were detected with an RF-20A fluorometer (Shimadzu), using excitation and emission wavelengths of 280 and 340 nm for the detection of intrinsic tryptophan fluorescence.

The residual ratio after incubation at each temperature was estimated using the fluorescence intensity at the elution peak, which corresponded to the T1rLBD dimer, that is, the peak height at ~11.6 min. The values were normalized to the intensity of the sample incubated at 4°C as 1. In order to estimate the apparent melting temperature ($T_{m\text{-app}}$) of the sample, the values of residual ratio at each temperature were fitted to the Gibbs–Helmholtz equation transformed as shown below (assuming that the sample protein is under equilibrium between a folding and unfolding state under each condition):

$$\text{Residual ratio} = 1 - \frac{1}{1 + exp\left[ -\frac{\Delta H\left(1 + \frac{T}{T_{m-app}}\right) - \Delta C_P\left\{ (T_{m-app} - T) + Tln\left(\frac{T}{T_{m-app}}\right)\right\}}{RT}\right]}$$

where $\Delta H$ and $\Delta C_p$ are the enthalpy and heat capacity change of unfolding, respectively; $T$ is the temperature of the sample incubation; $R$ is the gas constant. The fittings were performed with KaleidaGraph (Synergy Software), with $\Delta H$, $\Delta C_p$, and $T_{m\text{-app}}$ are set as valuables.

## Single fiber recording from mouse CT nerve

All animal experiments were conducted following the National Institutes of Health Guide for the Care and Use of Laboratory Animals and approved by the committee for Laboratory Animal Care and Use and the local ethics committee at Tokyo Dental College (Permit Number: 228101) and Okayama University (Permit Number: OKU-2022897) Japan. The subjects were six adult male C57BL/6JCrj mice (Charles River Japan, Tokyo, Japan) and four T1r3GFP-KO mice, which were obtained by mating T1r3-GFP (*Damak et al., 2008*) and T1r3-KO (*Damak et al., 2003*) mice. Mice were maintained on a 12/12 hr light/dark cycle and fed standard rodent chow and 8–20 weeks of age ranging in weight from 20 to 30 g.

The mice were anesthetized with an injection of combination anesthetic agents contained midazolam (0.8 ml/kg, Sandoz, Yamagata, Japan), medetomidine (0.75 ml/kg, Nippon Zenyaku Kogyo Co, Fukushima, Japan), butorphanol tartrate (1 ml/kg, Meiji Seika Pharma, Tokyo, Japan), and physiologic saline (7.45 ml/kg), and maintained at a surgical level of anesthesia, with additional injections of sodium pentobarbital (Nakarai Tesque, Kyoto, Japan, 8–10 mg/kg ip every hour). Under anesthesia, each mouse was fixed in the supine position with a head holder, and the trachea cannulated. The right CT nerve was dissected, free from surrounding tissues, after the removal of the pterygoid muscle and cut at the point of its entry to the tympanic bulla. A single or a few nerve fibers were teased apart with a pair of needles and lifted onto an Ag-AgCl electrode, and an indifferent electrode was placed in nearby tissue. Their neural activities were amplified (LP511; Grass Amplifier, Astro-Med, West Warwick, RI, USA) and recorded on a computer using a PowerLab system (PowerLab/sp4; AD Instruments, Bella Vista, NSW, Australia). For taste stimulation of fungiform papillae, the anterior half

of the tongue was enclosed in a flow chamber. Taste solutions or rinses (distilled water) (~24°C) were delivered to the tongue by gravity flow at the same flow rate (~0.1 ml/s). For data analysis, we used the net average frequency for 10 s after the stimulus onset, which was obtained by subtracting the spontaneous frequency for the 10-s duration before stimulation from after stimulation. In the initial survey to identify a nerve fiber connecting to T1r-expressing cells, test stimuli such as 100 mM NaCl, 10 mM HCl, 500 mM sucrose, 100 mM monopotassium glutamate, and 20 mM quinine HCl were separately applied. If the fiber responded to sucrose, we applied 10 µM to 100 mM NMDG-Cl, NaCl, KCl, or either of 20 mM L-glutamine or 100 mM sucrose with or without 10 mM NMDG-Cl to the tongue. The criteria for the occurrence of response were the following: the number of spikes was larger than the mean + two standard deviations of the spontaneous discharge for three 10-s periods before stimulation, and at least three spikes were evoked by taste stimulation (*Yasumatsu et al., 2012*). In the case of T1r3GFP-KO mice, as the mice showed a significant response to 0.5 M sucrose, we could identify sweet-responsive fibers (impulse frequency of 13.4 ± 1.31) (*Yasumatsu et al., 2012*). The reagents used were purchased from Wako Pure Chemical Industries (Osaka, Japan; others). To block responses via T1r (*Daly et al., 2013*; *Margolskee et al., 2007*; *Ninomiya and Imoto, 1995*; *Ninomiya et al., 1999*), each tongue was treated with 30 µg/ml (~7 µM) gurmarin (Gur) dissolved in 5 mM phosphate buffer (pH 6.8) for 10 min, similarly as described by *Ninomiya and Imoto, 1995*. To assure the detection of responses from T1r-expressing cells, recordings from Gur-insensitive sweet-responsive fibers were defined as those retaining impulse frequencies to 0.5 M sucrose more than 60% after the Gur treatment (*Ninomiya et al., 1999*) were excluded from the data. The number of Gur-sensitive and Gur-insensitive fibers was 6 and 3, respectively, among 0.5 M sucrose responding fibers. The sucrose application was repeated three to six times during the recordings. Additionally, the recovery of the suppressed responses was confirmed using 15 mM β-cyclodextrin, which could remove the effect of Gur from the tongue (*Ninomiya et al., 1999*). All Gur-sensitive fibers recovered up to 60–150% of responses before Gur. At the end of the experiment, animals were killed by administering an overdose of the anesthetic. Repeated measures ANOVA and Student's paired *t*-test were used to statistically evaluate the effects of chemicals or gene deletion. The sample size was calculated according to power analysis, thereby resulting in three per group due to the effect size (*d*) of 6–9.3 to detect the effects of blocking or deleting T1r3.

## Two-bottle preference tests

All training and testing sessions occurred during the light phase of the light/dark cycle. On the first day of training, the WT mice (adult male C57BL/6JCrj) were water deprived for 23 hr and then placed in a test box with two bottles: one filled with water and the other empty. The amount of fluid intake was measured after a 5-min presentation. After 4 days of training, mice were used for test sessions if they drank water evenly on either side (nine mice). In the test sessions, they were provided with two bottles, one containing 10 mM NMDG-Cl and the other containing water, with or without Gur, for 5 min. The amount of liquid consumed was measured by weighing the bottles, and a preference score of NMDG-Cl was calculated using the following equation:

$$\text{Preference score}\,(\%) = \frac{V_{Cl}}{V_{Cl}+V_w} \times 100$$

where $V_{Cl}$ and $V_w$ are the amount of NMDG-Cl intake and water intake, respectively.

## Acknowledgements

We thank Drs Kazuya Hasegawa, Nobuhiro Mizuno, and Naohiro Matsugaki for help with X-ray data collection; Junya Nitta and Hikaru Ishida for help with protein preparation; Ryusuke Yoshida for help with the single fiber recording; Yuko Kusakabe for attempt at cell-based receptor assay in the early stage of the study; Haruo Ogawa and Francine Acher for sharing knowledge about ANPR and mGluRs, respectively; Yuzo Ninomiya for valuable discussions. We also thank the reviewers of BioPhysics Colab for their helpful comments and Enago (https://www.enago.jp/) for the English language review. The synchrotron radiation experiments at the BL41XU, SPring-8 were performed with approvals of the Japan Synchrotron Radiation Research Institute (JASRI) (Proposal No. 2016B2534). The synchrotron radiation experiment at the BL-1A, Photon Factory was supported by the Platform for Drug Discovery, Informatics, and Structural Life Science (Proposal No. 1264).

This work was financially supported by JSPS KAKENHI Grant Numbers JP17H03644, JP18H04621, JP20H03195, JP20H04778, JP21H05524 (to AY) and JP20H03855, JP20K02415 (to KY), Mishima Kaiun Memorial Foundation, and the Salt Science Research Foundation (Proposal No. 2039) (to AY).

## Additional information

### Funding

| Funder | Grant reference number | Author |
|---|---|---|
| Japan Society for the Promotion of Science | JP17H03644 | Atsuko Yamashita |
| Japan Society for the Promotion of Science | JP18H04621 | Atsuko Yamashita |
| Japan Society for the Promotion of Science | JP20H03195 | Atsuko Yamashita |
| Japan Society for the Promotion of Science | JP20H04778 | Atsuko Yamashita |
| Japan Society for the Promotion of Science | JP21H05524 | Atsuko Yamashita |
| Japan Society for the Promotion of Science | JP20H03855 | Keiko Yasumatsu |
| Japan Society for the Promotion of Science | JP20K02415 | Keiko Yasumatsu |
| Mishima Kaiun Memorial Foundation | | Atsuko Yamashita |
| Salt Science Research Foundation | 2039 | Atsuko Yamashita |

The funders had no role in study design, data collection, and interpretation, or the decision to submit the work for publication.

### Author contributions

Nanako Atsumi, Conceptualization, Formal analysis, Investigation, Writing – review and editing; Keiko Yasumatsu, Conceptualization, Formal analysis, Funding acquisition, Investigation, Writing – original draft, Writing – review and editing; Yuriko Takashina, Chiaki Ito, Formal analysis, Investigation, Writing – review and editing; Norihisa Yasui, Investigation, Writing – review and editing; Robert F Margolskee, Resources, Writing – review and editing; Atsuko Yamashita, Conceptualization, Data curation, Formal analysis, Supervision, Funding acquisition, Investigation, Writing – original draft, Writing – review and editing

### Author ORCIDs

Keiko Yasumatsu http://orcid.org/0000-0002-1911-8792
Norihisa Yasui http://orcid.org/0000-0001-7117-3070
Robert F Margolskee http://orcid.org/0000-0002-9572-2887
Atsuko Yamashita http://orcid.org/0000-0002-8001-4642

### Ethics

All animal experiments were conducted following the National Institutes of Health Guide for the Care and Use of Laboratory Animals and approved by the committee for Laboratory Animal Care and Use and the local ethics committee at Tokyo Dental College (Permit Number: 228101) and Okayama University (Permit Number: OKU-2022897) Japan.

### Decision letter and Author response

Decision letter https://doi.org/10.7554/eLife.84291.sa1
Author response https://doi.org/10.7554/eLife.84291.sa2

## Additional files

### Supplementary files
• MDAR checklist

### Data availability
All data generated or analyzed during this study are included in the manuscript and supporting file; Source Data files have been provided for Figures 2, 3, and 4.

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
