## [Editor Report]

The manuscript by Atsumi et al. presents solid evidence that identifies the T1r (sweet /umami) taste receptors as chloride (Cl^-^) receptors. The authors employ many state-of-the-art techniques to demonstrate that T1r receptors from Medaka fish bind chloride and that this binding induces a conformational change in the heteromeric receptor. This conformational change leads to low-concentration chloride-specific action potential firing in nerves from neurons containing these receptors in mice. These results represent an important advance in our understanding of the logic of taste perception.

---

## [Decision Letter]

**Decision letter after peer review:**

Thank you for submitting your article "Chloride ions evoke taste sensations by binding to the extracellular ligand-binding domain of sweet/umami taste receptors" for consideration by *eLife*. Your article has been reviewed by Leon D Islas as Reviewing Editor and Reviewer #1, and the evaluation has been overseen by Richard Aldrich as the Senior Editor.

The Reviewing Editor has drafted this to help you prepare a revised submission.

Essential revisions:

The authors have satisfactorily responded to the comments raised by a review of their Preprint from Biophysics Colab, and have received an Endorsement. There are two remaining points that need to be dealt with in the discussion of a revised submission.

1) The FRET measurements employ as a donor the Venus fluorescent protein, which is a variant of YFP, whose fluorescence intensity is highly chloride dependent. While the Kd for chloride of YFP is high and possibly higher than the chloride concentrations employed for FRET experiments, the authors should discuss the possibility of artifacts in their FRET experiments.

2) While the evidence indicating increased excitability by chloride sensing via T1r receptors is solid, there is no discussion of the possible signaling pathways involved. The authors should feel free to propose, based on what is known about T1r -containing neurons, possible transduction pathways leading from Cl^-^ binding to T1r to the generation of action potentials.

---

## [Author Response]

Essential revisions:The authors have satisfactorily responded to the comments raised by a review of their Preprint from Biophysics Colab, and have received an Endorsement. There are two remaining points that need to be dealt with in the discussion of a revised submission.1) The FRET measurements employ as a donor the Venus fluorescent protein, which is a variant of YFP, whose fluorescence intensity is highly chloride dependent. While the Kd for chloride of YFP is high and possibly higher than the chloride concentrations employed for FRET experiments, the authors should discuss the possibility of artifacts in their FRET experiments.

In this study, we used Venus, a YFP variant with much lower halide sensitivity (with *K*_d_ for Cl^–^ is >10^4^ mM) than the original YFP, as a FRET acceptor. Therefore, we believe that the Cl^–^ effects on Venus in the Cl^–^ titration experiment (up to 10 mM) are negligible. This notion was added to the subsection entitled “Förster resonance energy transfer analysis” in the Materials and methods section, as follows.

“While the yellow fluorescent protein, a commonly used FRET acceptor, is sensitive to halides (Wachter and Remington, 1999), Venus is a halide-insensitive variant with the *K*_d_ value for Cl^–^ as >10^4^ mM (Nagai et al., 2002). Therefore, fluorescence changes due to chloride ions themselves, which are not due to energy transfer of the fluorescent proteins, are most likely negligible under the conditions tested in this study.”

2) While the evidence indicating increased excitability by chloride sensing via T1r receptors is solid, there is no discussion of the possible signaling pathways involved. The authors should feel free to propose, based on what is known about T1r -containing neurons, possible transduction pathways leading from Cl^-^ binding to T1r to the generation of action potentials.

We appreciate the reviewer’s valuable suggestion. We added the following description in the Discussion section.

“The Cl^–^-induced conformational change is similar to that induced by canonical taste substances for T1rs, amino acids, though its efficacy is slightly lower. Therefore, as with other class C GPCRs, the structural change at LBD caused by Cl^–^-binding most likely provokes receptor activation, resulting in G-protein activation in taste cells. The signal is considered to be transmitted further through the common downstream cascade of T1rs: activation of phospholipase Cβ_2_ and resultant inositol triphosphate (IP_3_) production, IP_3_-dependent activation of IP_3_ receptors followed by ca^2+^-release from endoplasmic reticulum, ca^2+^-dependent activation of TRPM5 channel inducing taste cell depolarization, and subsequent action potential generation and neurotransmitter ATP release through CALHM1/3 (Taruno et al., 2021).”